# How to Improve Solubility and Dissolution of Irbesartan by Fabricating Ternary Solid Dispersions: Optimization and In-Vitro Characterization

**DOI:** 10.3390/pharmaceutics14112264

**Published:** 2022-10-23

**Authors:** Aasma Akram, Muhammad Irfan, Walaa A. Abualsunun, Deena M. Bukhary, Mohammed Alissa

**Affiliations:** 1Department of Pharmaceutics, Faculty of Pharmaceutical Sciences, Government College University, Faisalabad 38000, Pakistan; 2Department of Pharmaceutics, Faculty of Pharmacy, King Abdulaziz University, Jeddah 21589, Saudi Arabia; 3Department of Pharmaceutics, College of Pharmacy, Umm Al-Qura University, Makkah 24381, Saudi Arabia; 4Department of Medical Laboratory Sciences, College of Applied Medical Sciences, Prince Sattam bin Abdulaziz University, Al-Kharj 11942, Saudi Arabia

**Keywords:** health care, solid dispersion, Irbesartan, Soluplus^®^, Kollidon^®^ VA 64, Kolliphor^®^ P 407, Polyinylpyrrolidone (PVP-K30), solubility, dissolution

## Abstract

The purpose of this study is to improve the solubility and dissolution of a poorly soluble drug, Irbesartan, using solid dispersion techniques. For that purpose, different polymers such as Soluplus*^®^,* Kollidon^®^ VA 64, Kolliphor^®^ P 407, and Polyinylpyrrolidone (PVP-K30) were used as carriers at different concentrations to prepare solid dispersion formulations through the solvent evaporation method. In order to prepare binary dispersion formulations, Soluplus^®^ and Kollidon^®^ VA 64 were used at drug: polymer ratios of 1:1, 1:2, 1:3, and 1:4 (*w/w*). Saturation solubility of the drug in the presence of used carriers was performed to investigate the quantitative increase in solubility. Dissolution studies were performed to explore the drug release behavior from the prepared dispersions. Additionally, the characterization of the prepared formulations was carried out by performing DSC, SEM, XRD, and FTIR studies. The results revealed that among binary systems, K_4_ formulation (Drug: Kollidon^®^ VA 64 at ratio of 1:4 *w/w)* exhibited optimal performance in terms of increased solubility, drug release, and other investigated parameters. Furthermore, ternary dispersion formulations of the optimized binary formulation were prepared with two more polymers, Kolliphor^®^ P 407 and Polyvinylpyrrolidone (PVP-K30), at (Drug: Kollidon^®^ VA 64:ternary polymer) ratios of 1:4:1, 1:4:2, and 1:4:3 (*w/w*). The results showed that KPVP (TD_3_) exhibited the highest increase in solubility, as well as dissolution rate, among ternary solid dispersion formulations. Results of solubility enhancement by ternary solid dispersion formulations were also supported by FTIR, DSC, XRD, and SEM analysis. Conclusively, it was found that the ternary solid dispersion-based systems were more effective compared to the binary combinations in improving solubility as well as dissolution of a poorly soluble drug (Irbesartan).

## 1. Introduction

High-throughput screening techniques and combinatorial chemistry have discovered many new compounds as therapeutic agents that possess poor solubility. In recent years, 40–70% of newly discovered drugs have been found to be poorly soluble [1]. A majority of these newly discovered compounds are abandoned at the beginning of the drug development process due to their very low solubility. Therefore, there is a need to overcome these poor solubility issues in order to improve bioavailability and therapeutic effect [2].

A majority of drugs are administered orally as this route provides the easiest method of administration and maximum patient compliance, however, this route also presents many issues. Poor bioavailability due to low solubility of a drug is one of the major hurdles involved in drug administration through the oral route. There are a variety of factors that hinder absorption of drugs into the gastrointestinal tract (GIT), with low water solubility and/or low permeability of the substance as major factors. GIT is a long tubular tract representing the set of organs that extends from mouth to anus, but usually refers to the stomach and intestine [3]. For absorption through the oral route, drugs must first dissolve in GIT fluid before permeating through GIT membranes to reach the blood. Therefore, a drug with low solubility will show a dissolution rate with limited absorption. Consequently, there is a need to enhance the solubility of such compounds/substances to increase their bioavailability [4].

Commonly available methods for solubility enhancement are formation of salt, use of solubilizers, reduction in particle size, complex formation, and solid dispersion. All of these techniques have their own advantages and disadvantages. The effect of common ions, instability of the salt form, or not achieving the targeted solubility are the major possible disadvantages of the salt formation method, whereas the limitations of the particle size reduction technique include agglomeration of particles. In contrast, solid dispersion has great potential for increasing solubility as this inhibits aggregation of particles. In addition to inhibition of particle aggregation, solid dispersion also reduces solid liquid surface tension [5].

Amorphous solid dispersions are defined as the dispersion of one or more APIs (hydrophobic) in a polymeric matrix (hydrophilic) at a solid state [6]. Improved wettability, decreased particle size, inhibition of aggregation, and conversion of the drug from crystalline to amorphous phase are important mechanisms involved in enhancing solubility [7].

Polymers selected for this study are hydrophilic and commonly used for improving solubility. Soluplus^®^ is a biodegradable and biocompatible polymer [8] composed of polyethylene glycol, polyvinyl acetate, and polyvinyl caprolactam and is amphiphilic in nature having Tg of 70 °C [9]. The molecular weight of Soluplus^®^ is in the range of 90–140,000 g/mol. It contains PEG 6000 as a backbone linked with one or two side chains composed of vinyl acetate, which is randomly copolymerized with vinyl caprolactam [10]. This polymer possess a hydrogen bond acceptor group (1-methyl-2-pyrrolidone) which is responsible for hydrogen bonding [11]. Kollidon^®^ VA 64 is another hydrophilic polymer composed of two monomers, N-vinylpyrrolidone and vinyl acetate, linked in a ratio of 6:4 by free radical polymerization. This molar ratio is represented as 64 in the trade name of this polymer [12]. Kollidon^®^ VA 64 has two hydrogen bond acceptor groups (from the carbonyl group of the pyrrolidone ring and vinyl acetate) [13]. Kolliphor^®^ P 407 is a triblock polymer composed of a central block of hydrophobic polypropylene oxide (PPO) surrounded by two hydrophilic blocks of polyethyleneoxide (PEO). The molecular weight of this polymer is in the range of 9840–14600 daltons [14,15]. It shows the phenomena of micellization due to its amphiphilic nature [16]. PVP, also named as povidone, is a water-soluble, nontoxic, biocompatible polymer made up of linear 1-vinyl-2-pyrrolidone groups [2]. The approximate molecular weight of this polymer is 50,000 daltons [17]. This polymer also possess a hydrogen bond acceptor group which is a pyrrolidone ring [11]. The structures of the polymers are given in Figure 1B–E.

Irbesartan (Irb) is an angiotensin II (AT1 receptor) inhibitor which is used to treat hypertension alone or in combination with other drugs. The initial dose to treat hypertension is 150 mg once daily, but in cases of severe disease this dose can be raised to 300 mg once daily [18]. The molecular weight of Irbesartan is 428.5 g/mol. The log P value of this drug is 10.1 and its quantitative solubility in water is 5.9 × 10^−2^ mg/L at 25 °C. Its structure is provided in Figure 1A. This low solubility is a challenge in the drug development process as it leads to a poor drug release profile and low therapeutic effect [19]. The solid dispersion technique is used to overcome these challenges of poor solubility and bioavailability. In this work, binary and ternary solid dispersion-based formulations of Irbesartan were fabricated through the solvent evaporation method to investigate the effect on solubility as well as dissolution. The prepared solid dispersion formulations were further characterized for FTIR, DSC, XRD, and SEM studies.

## 2. Materials and Methods

### 2.1. Materials

Irbesartan was generously given by Barrett Hodgson Pvt., Ltd., Karachi, Pakistan. Soluplus^®^*,* Kollidon^®^ VA 64, and Kolliphor^®^ P 407 were gifted by BASF Pharma., Ludwigshafen, Germany. Polyvinylpyrrolidone (PVP-K30) and methanol were bought from Daejung Chemicals and Metals Co., Ltd., Korea. The remaining chemicals used were of analytical grade.

### 2.2. Screening Study

A phase solubility, or screening study, was conducted to select the most suitable carrier for solid dispersions. Polymers included in the phase solubility study were Soluplus^®^*,* Kollidon^®^ VA 64, Kolliphor^®^ P 407, Polyinylpyrrolidone (PVP-K30), PEG 6000, HPMC E5, and Gelucire^®^ 50/13.

Polymeric solutions of various concentrations (1%, 2%, 3%, and 4% *w/v*) were formulated in distilled water and supersaturated solutions of these solutes were prepared by adding an excess amount of drug to each falcon tube. These falcon tubes containing supersaturated solutions were positioned in a shaker at 37 °C for 72 h. After centrifugation (4000 rpm for 20 min), these samples were filtered using a 0.45 µm syringe filter. After appropriate dilution, absorbance of each sample was taken at 253 nm using a UV spectrophotometer. Gelucire^®^ 50/13 solubility was taken at lower concentrations due to its insolubility in water at higher concentration [20].

### 2.3. Preparation of Solid Dispersion Formulations

Solid dispersion-based formulations were prepared using the solvent evaporation method. For binary solid dispersions, two hydrophilic polymers, Soluplus^®^ and Kollidon^®^ VA 64, were used at ratios (Drug: Polymer) of 1:1, 1:2, 1:3, and 1:4 (*w*/*w*). Ternary solid dispersions of optimized binary formulation were prepared with two more polymers, Kolliphor^®^ P 407 and Polyvinylpyrrolidone (PVPK-30). Ternary dispersions were prepared at ratios (Drug: Kollidon^®^ VA 64: ternary polymer) of 1:4:1,1:4:2,1:4:3 (*w*/*w*/*w*). Formulae of binary and ternary solid dispersions are given in Table 1.

The appropriate quantity of Irbesartan (approx. 1 g) was dissolved in adequate volume (20 mL) of methanol. The selected carrier was also dissolved in 20 mL methanol separately under constant stirring. These two solutions were mixed together by sonication for 10 min in order to have uniform mixing. Solvent evaporation was carried out using a rotary evaporating apparatus (RE-100 Pro, Scilogex, CT, USA) at 45 °C. The product obtained was dried at 40 °C for 24 h. After pulverization, the dried product was passed through a sieve of mesh No. 80. The dried powder was stored in a tightly closed bottle with desiccating agents for future use [21]. For ternary solid dispersion, prepared binary solid dispersion was dissolved in methanol (30 mL) and the selected carrier was also dissolved in 20 mL methanol separately. These two solutions were mixed together with the help of sonication. Further preparation steps were the same as for binary solid dispersion preparation. Steps involved in preparation of binary and ternary solid dispersions are presented diagrammatically in Figure 2.

### 2.4. Solubility Studies

Solubility was determined by adding pure drug and solid dispersion formulations separately in falcon tubes containing either 0.1 N HCl (1.2 pH) or a phosphate buffer of pH 6.8. These samples were vortexed to prepare supersaturated solution and then placed in a shaker (SWB 15, Thermo-scientific, Waltham, MA, USA) for 72 h at 37 °C. Then, centrifugation was performed at speed of 5000 rpm for 20 min. After proper dilution and filtration of supernatant, analysis was carried out with the help of a UV–visible spectrophotometer (CE-7400S, Cecil, Cambridge, UK) at 253 nm [22].

### 2.5. Dissolution Studies

Dissolution behavior of pure drug and solid dispersion formulations were studied using USP apparatus II (DT 70, Pharma Test, Hainburg, Germany). This study was conducted using 900 mL of 0.1 N HCl (pH 1.2) and a phosphate buffer of pH 6.8 at 37 °C with stirring speed 100 rpm. Solid dispersion containing 10 mg of API was placed in each dissolution vessel. A 5.0 mL sample aliquot was taken at different time intervals (0, 5, 15, 30, 45, 60, 90, and 120 min through a 0.45 µm Millipore filter. After withdrawal of a sample aliquot, an equivalent amount of freshly prepared dissolution medium was added to maintain sink conditions. Analysis was carried out with a UV–visible spectrophotometer (CE-7400S, Cecil, Cambridge, UK) at 253 nm [21].

### 2.6. Drug Content and Yield

To determine drug content, an amount of solid dispersion containing 10 mg of drug substance was added to 10 mL of solvent (methanol) and sonicated until the formulation was completely dissolved. After filtration, the UV–visible spectrophotometry technique was used at 253 nm (lambda max) and the amount of drug was determined from the calibration curve.

The following formula was used to calculate % drug contents:% Drug content = calculated drug content/theoretical drug content × 100

In order to determine % yield, there was need to determine the weight of dried solid dispersions (W1) obtained from prepared batches and the initial weight of drug and polymer/s (W2).

The following formula was used to calculate % yield:% Yield = W1/W2 × 100

### 2.7. Kinetic Model Analysis

Drug release data was fitted into various release models in order to understand the release mechanism using DD solver software.

The equation for zero order is given below. This equation is used to describe concentration independent drug release.
C = k_o_·t
where, k_o_ = zero-order rate constant and t = time.

The equation for first order is
Log C = LogC_o_ − Kt/2.303
where, C_o_ = initial concentration of drug and K = first order constant. The first order describes concentration-dependent drug release.

The equation for Higuchi model is
Q = kt_1/2_
where Q = percentage of drug released at time (t) and k = Higuchi coefficient

The Hixson–Crowell model equation is
Qo^1/3^ − Qt^1/3^ = k_HC_·t
where is Qt = Amount of drug released in time t, Qo = Initial concentration of drug in any dosage form, k_HC_ = Hixson–Crowell rate constant.

The equation for Korsmeyer–peppas model is
Q = k_p_t^n^
where Q = percent of the drug released at time (t), k_p_ is Peppas constant, and n is the release exponent which is used to understand the release mechanism [23] (Table 2).

### 2.8. Scanning Electron Microscopy (SEM)

SEM was performed to reveal the surface characteristics of pure Irbesartan and solid dispersion samples by using an electron microscope (Nova-Nano-450, FEI, Hillsboro, OR, USA). Adhesive tape was used to mount the prepared samples onto the carbon tail ends, after which they were coated with an alloy made up of gold and palladium. Atmospheric pressure was maintained at 0.25 Torr using an ion sputter coater and finally the samples were placed in the scanning chamber [24].

### 2.9. Powder X-ray Diffraction (PXRD)

Changes in drug crystalline structure were determined using an X-ray Diffractometer (JDX3522, Japan). PXRD was carried out by placing samples in an X-ray diffraction device and exposing them to varied angles (4° to 50°) 2θ with scanning speed maintained at 1°/min [25].

### 2.10. FTIR Analysis

FTIR spectroscopy studies were performed using an FTIR spectrophotometer (Agilent scientific Instruments, Santa Clara, CA, USA) to identify drug substance and to characterize major interactions between API and polymers. Samples were mixed with potassium bromide in a glass mortar and then pressed to prepare potassium bromide discs. Scans were run from 500 to 4000 cm^−1^ with a resolution power of 1 cm^−1^ [26].

### 2.11. Differential Scanning Calorimetry (DSC)

DSC thermograms of drug, polymers, and formulations were obtained using a thermal analyzer (SDT Q600, V20.9 Build 20, TA instruments, New Castle, DE, USA). An amount of 5 mg of test substance was kept on a DSC pan made up of aluminum. The temperature to start the test was 25 °C. The temperature was raised by maintaining a speed of 10 °C/min, with a temperature upper limit of 350 °C or T_m_ of the substance. Nitrogen gas flow was maintained at a speed of 30 mL/min [27].

## 3. Results and Discussion

### 3.1. Screening Study

All hydrophilic polymers showed an increase in solubility compared to pure drug in the screening study. The results of the screening study are shown in Table 3. Based on these results, two polymers, namely Soluplus^®^ and Kollidon^®^ VA 64, were selected to prepare binary solid dispersions, as these polymers showed the highest increase in solubility.

### 3.2. Solubility Studies

The solubility of pure drug in distilled water, pH 6.8 phosphate buffer, and 0.1 N HCl (pH 1.2) was 10.84 ± 1.079 µg/mL, 19.78 ± 1.56 µg/mL, and 30.72 ± 2.59 µg/mL, respectively. The results of the solubility studies of the prepared formulations are represented in Figure 3. They revealed that the K_4_ formulation exhibited the highest saturation solubility (754.08 ± 1.70 µg/mL in 0.1 N HCl and 540.97 ± 1.57 µg/mL in pH 6.8 phosphate buffer) in the case of binary solid dispersions. However, KPVP (TD_3_) ternary solid dispersion was found to have a saturation solubility of 820.93 ± 3.26 µg/mL and 924.71 ± 0.98 µg/mL in pH 6.8 phosphate buffer and 0.1 N HCl, respectively, which was the maximum increase in solubility among all the prepared ternary solid dispersions. This maximum increase in solubility could be linked to the complete transformation of the crystalline form of the drug into its amorphous form in KPVP (TD_3_) formulation.

### 3.3. Dissolution Studies

The results revealed that the pure drug showed 46.03 ± 0.141% release in 0.1 N HCl and 22.31 ± 0.169% release in pH 6.8 phosphate buffer after 2 h. It was noticed that during first 30 min, the drug release was 11.00 ± 1.49% in 6.8 pH phosphate buffer and 36.55 ± 0.915% in 0.1 N HCl. Overall, all the prepared formulations showed an increase in drug release compared to pure drug. This could be attributed to the transformation of the crystalline drug into its amorphous form, reduced particle size, and increased wetting [22,28]. In addition, Soluplus^®^-based binary solid dispersions showed increased drug release, probably due to the maintenance of supersaturation which might result from intermolecular bonding between API and Soluplus^®^ (Figure 4). Noticeably, Soluplus^®^-based solid dispersions provided decreased drug release with increasing Soluplus^®^ concentration. This may be due to the fact that Soluplus^®^ possesses swelling properties, which may decrease drug release due to the limited diffusion of drug through the swelled polymer [29,30]. The binary system prepared with Kollidon^®^ VA 64 (K_4_) was further selected for the preparation of ternary system as it provided the highest drug release of 71.21 ± 0.364% and 79.75 ± 0.094% in pH 6.8 phosphate buffer and 0.1 N HCl (pH 1.2) after 120 min, respectively (Figure 5). This maximum increase in solubility and dissolution rate might be associated with sustained supersaturation, formation of nanoparticles during dissolution, crystal nucleation, and growth inhibition by Kollidon^®^ VA 64 (polymer shows adsorption on drug particles). Another possible mechanism involved in enhancing the solubility of this polymer could be the formation of intermolecular bonding, especially hydrogen bonding between carbonyl functional group of pyrrolidone and vinyl acetate [29]

In the case of ternary systems, the formulation KP407 (TD_3_) exhibited drug release of 82.46 ± 0.082% and 93.20 ± 0.076% in pH 6.8 phosphate buffer and 0.1 N HCl (pH 1.2), respectively, after 120 min, and the formulation KPVP (TD_3_) showed drug release of 87.31 ± 0.805% and 97.60 ± 0.887% in pH 6.8 phosphate buffer and 0.1 N HCl, respectively, after 120 min. The increase in solubility and drug release in the case of Kolliphor^®^ P 407-based ternary dispersion might be linked to the surfactant nature of this polymer (Figure 6). Kolliphor^®^ P 407 is composed of ethylene oxide and propylene oxide blocks, which are responsible for self-assembly into micelles in aqueous media, thus leading to solubilization of the drug. The mechanism by which PVP-K30 increased drug release is most likely the maintenance of supersaturation resulting from the crystal growth inhibition potential of PVP. Hydrogen bonding between PVP and the weakly basic drug may be another reason for the increased solubility as well as dissolution [29,31].

Based on the results of solubility and dissolution studies, KPVP (TD_3_) ternary solid dispersion formulation was considered as the optimized formulation (Figure 7).

### 3.4. Drug Content and % Yield

The results for drug content and percent yield are presented in Table 4. All the formulations provided more than 80% drug content and yield. Drug content and yield of all formulations ranged from 81.20 ± 0.011% to 94.56 ± 0.005% and 81.20 ± 1.085% to 94.33 ± 1.258%, respectively. Importantly, the optimized formulation KPVP (TD3) revealed 94.56 ± 0.005% drug content and 94.33 ± 1.258% yield.

### 3.5. Kinetic Modelling

The results of kinetic modelling analysis are shown in Table 5 and Table 6. All solid dispersion formulations followed the Korsmeyer–Peppas model, as the values of R^2^ for this model were higher compared to other models. Moreover, the values of n obtained from the slope of the Korsemeyer–Peppas model for all solid dispersions were less than 0.45 which shows that the drug release mechanism followed Fickian diffusion [32]. A possible explanation for this mechanism may be that the hydrophilic polymer formed a diffusion layer surrounding the drug particles leading to an alteration in the hydrophobicity of drug, increased wetting, and reduced particle size and crystallinity. Therefore, the drug had to pass through this diffusion layer in order to come in contact with dissolution medium. It is also reported from the literature that formation of such a diffusion layer by hydrophilic polymers supports this type of release behavior [33].

### 3.6. SEM

The SEM micrographs of pure Irbesartan and of the prepared formulations S_1_, K_4_, KP407 (TD_3_), and KPVP (TD_3_) are shown in Figure 8. It was found that the SEM micrograph of Irbesartan clearly exhibited needle type lumps or crystals of pure drug, thus confirming its crystalline nature [22].

S1 and K_4_ formulations showed irregular amorphous forms of the drug with few needles. However, the SEM analysis of ternary dispersion KP407 (TD_3_) confirmed almost complete transformation of crystalline API into amorphous form. In addition, the SEM micrographs of ternary dispersion KPVP (TD_3_) showed disappearance of crystals of drug due to its masking by the polymeric carriers. These SEM results were also supported by the PXRD finding confirming the presence of the amorphous form of the drug in the optimized solid dispersion formulation [34].

### 3.7. PXRD Studies

The PXRD data of pure Irbesartan and other samples are provided in Figure 9. The PXRD of Irbesartan showed intense peaks at 12.5°, 13.3°, 17.1°, 19.4°, 21.2°, 22.6°, 23.2°, 23.6°, and 27.3° [22] corresponding to crystalline form A of Irbesartan [35], whereas the Soluplus^®^ did not show any intense peaks due to its amorphous nature [36]. In comparison, the PXRD of solid dispersion (S_1_) with 1:1 ratio (drug: polymer) exhibited drug peaks with less intensity, reflecting its partial transformation into amorphous form [37]. Kollidon^®^ VA 64 showed two halos near 12 and 22 2θ [38]. The PXRD of formulation (K_4_) with 1:4 ratio (drug: polymer) exhibited only a few drug peaks with much less intensity, thus supporting more conversion into its amorphous form [16].

Kolliphor^®^ P 407 showed two peaks at 19.346° and 23.472° angles (2 θ) [16]. Polyvinylpyrrolidone-K30 did not show these typical diffraction peaks due to its amorphous nature [39], indicating a broad and diffused pattern as its molecules are randomly arranged in a crystal lattice [40]. Furthermore, the PXRD of formulation KP407 (TD_3_) with 1:4:3 ratio (drug: binary polymer:ternary polymer) revealed only one small peak, reflecting a high conversion of crystalline drug into amorphous form. Also, the PXRD of formulation KPVP (TD_3_) with 1:4:3 ratio (drug: binary polymer:ternary polymer) exhibited complete transformation of the drug into amorphous form, showing no peak.

### 3.8. FTIR Spectroscopy Studies

The FTIR spectra of pure Irbesartan provided peaks at 1610 cm^−1^ due to C-N stretch, 1727 cm^−1^ due to C-O stretch, 2900 cm^−1^ and 2964 cm^– 1^ due to N-H stretches, 1436 cm^– 1^ and 1409 due to NNH bending, and 778 cm^−1^ due to NH out of plane bending of the CNH group, which are in accordance with the previous literature [22,35,41] (Figure 10A). Peaks at 1727 cm^−1^, 1610 cm^−1^, 778 cm^−1^, 1517 cm^−1^, and 756 cm^−1^ corresponds to crystalline form A of Irbesartan [35,42]. The FTIR spectra of Soluplus^®^ showed peaks similar to those previously reported in the literature. Peaks seemingly due to O-H stretch appeared at 3460 cm^−1^, C-H stretch (aromatic) at 2915 cm^−1^, C=O stretch due to vinyl acetate at 1730 cm^−1^, C=O stretch due to vinyl caprolactam carbonyl at 1630 cm^−1^, and C-O-C stretch at 1470 cm^−1^ [43].

The prominent peaks of the API and the carrier were present in solid dispersion (S_1_) with lower intensity, which is indicative of no major interaction between drug and polymer. The presence of less intense and broadened peaks of drug at 2900 cm^−1^ and 2964 cm^– 1^ might be due to possible bonding between the N-H group of the drug and the carbonyl (C=O) group of Soluplus^®^ [43].

The peak of drug at 1727 cm^−1^ appears with less intensity and the peak at 1610 cm^−1^ is broadened in S_1_ formulation. This may be due to overlapping of the drug with the polymer.

Kollidon^®^ VA 64 possess two hydrogen bond acceptor groups which are from the carbonyl group of the pyrrolidone ring (at 1660 cm^−1^) and vinyl acetate (at 1737 cm^−1^) [13] (Figure 10B). Kollidon^®^ VA 64 also showed peaks at 3500 cm^−1^ and 2900 cm^−1^, probably due to a higher number of O-H groups [44]. Moreover, the major peaks of Irbesartan and Kollidon^®^ VA 64 were retained in formulation (K_4_), showing the compatibility of the drug and polymers. Peaks that appeared at 2900 cm^−1^ and 2964 cm^– 1^ were present in the formulation with less intensity. The drug peak at 1727 cm^−1^ in the formulation (K_4_) was less intense compared to its appearance in the spectrum of pure drug. The drug peak that appeared at 1610 cm^−1^ in FTIR was shifted to the higher frequency of 1666 cm^– 1^. Shifting towards higher wavelengths may be due to the presence of weak Van der Waal forces between the drug and polymer [22]. This broadening of peaks was also linked to the change of drug form (crystalline to amorphous) [45]. Proton donors, such as OH, and proton acceptors are present in Kollidon^®^ VA 64, which might have facilitated intermolecular interaction at these sites resulting in broadening of the peaks [46]. Major peaks of PVP-K30 appeared at 2940 (C–H), 1650 (C=O stretch), 1506 (C=C stretch), and 1273 cm^–1^ (C–N stretch) [47]. In optimized formulation KPVP (TD_3_), drug peaks appeared at 1610 cm^−1^ and 1727 cm^−1^ with less intensity, and peaks were broadened at 2900 cm^−1^ and 2964 with much less intensity, which could be due to hydrogen bonding between the carbonyl functional group of PVP and the N-H functional group of the drug [48].

In addition, Kolliphor^®^ P 407 showed peaks at 2893.02 cm^–1^ (C-H aliphatic stretching), 1342 cm^–1^ (O-H bending), and 1110 cm^–1^ (C-O stretching) [49]. In optimized KP407 (TD_3_) formulation, drug peaks were broadened at 1610 cm^–1^, 1732 cm^–1^, 2900 cm^–1^, and 2964 cm^–1^, with less intensity in ternary dispersion. This broadening and reduction in intensity clearly reflected reduced crystallinity of the drug in the formulation [50] (Figure 11).

### 3.9. DSC Studies

The DSC thermograms of pure drug, polymers, and formulations are shown in Figure 12. Pure Irbesartan depicted an intense endothermic peak at 190 °C due to its melting point which was shifted in solid dispersion formulations, indicating reduced crystallinity of pure drug in formulations [50,51]. The presence of a broader peak, or the complete absence of a melting peak, is a clear indication that a drug is partially or completely dispersed in polymeric carriers [52]. In contrary, Soluplus^®^ did not show any melting endothermic peak due to its amorphous nature [9]. It was noticed that the drug peak was broadened (170–184 °C) in the DSC curve of solid dispersion (S_1_) reflecting its partial dispersion in polymeric carrier. Furthermore, Kollidon^®^ VA 64 did not reveal any sharp melting endothermic peak due to its amorphous nature [44]. In the DSC curve of formulation (K_4_), the drug peak was shifted towards lower temperature (110–140 °C), supporting the partial conversion of the drug into its amorphous form. These results are in accordance with FTIR and XRD results.

Kolliphor^®^ P 407 exhibited an endothermic peak at 56 °C [53]. The thermogram of KP407 (TD_3_) formulation showed complete absence of a drug melting peak. PVP K30 provided a broad endothermic peak (70–130 °C) due to its amorphous nature. However, the DSC curve of KPVP (TD_3_) revealed a complete disappearance of drug peak, supporting its conversion into amorphous form [54].

## 4. Conclusions

In this research, binary and ternary solid dispersion-based formulations of Irbesartan were successfully formulated and evaluated. Clearly, the results showed that all the prepared formulations provided a significant increase in solubility as well as dissolution, however, the performance of ternary solid dispersions were potentially more effective when compared to binary solid dispersions. Moreover, the characterization data of FTIR, DSC, SEM, and XRD studies revealed the amorphous nature of the drug in the prepared solid dispersions. Finally, it was concluded that the prepared solid dispersion formulations could be potentially used to increase the solubility as well as dissolution of poorly soluble Irbesartan.

## Figures and Tables

**Figure 1 pharmaceutics-14-02264-f001:**
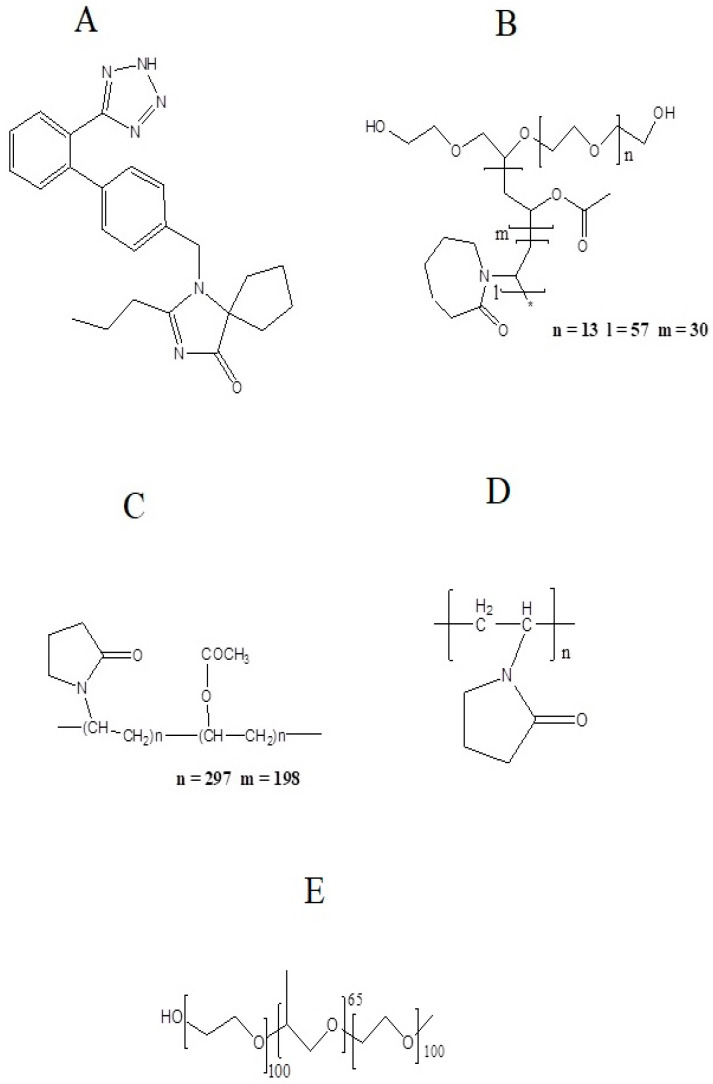
Structures of (**A**) Irbesartan, (**B**) Soluplus^®^, (**C**) Kollidon^®^ VA 64, (**D**) PVP-K30, and (**E**) Kolliphor^®^ P 407.

**Figure 2 pharmaceutics-14-02264-f002:**
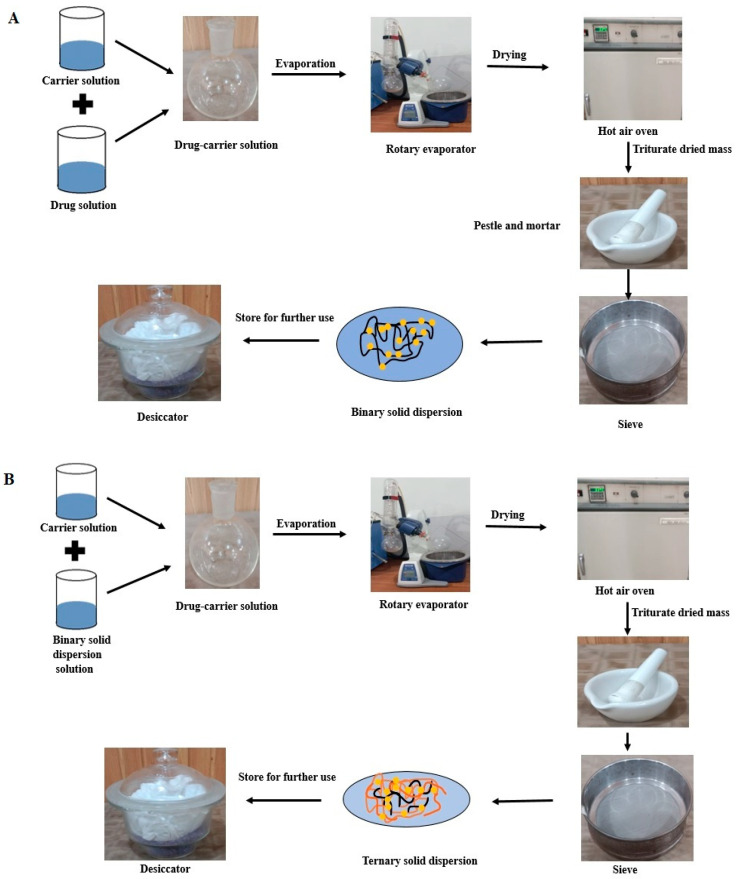
(**A**) Preparation of binary solid dispersions, (**B**) Preparation of ternary solid dispersions.

**Figure 3 pharmaceutics-14-02264-f003:**
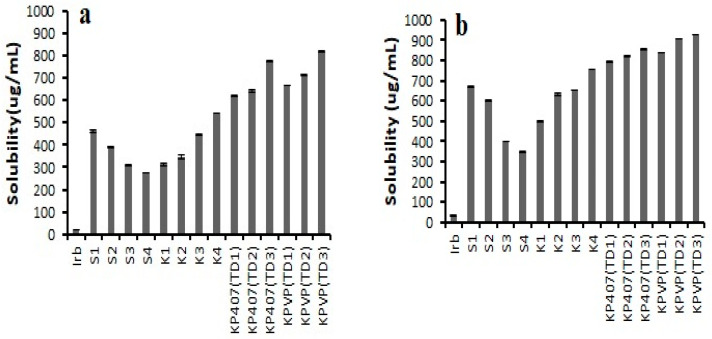
Saturation solubility data of pure Irbesartan and solid dispersions in (**a**) pH 6.8 phosphate buffer, (**b**) 0.1 N HCl (pH 1.2) (n = 3, mean ± SD).

**Figure 4 pharmaceutics-14-02264-f004:**
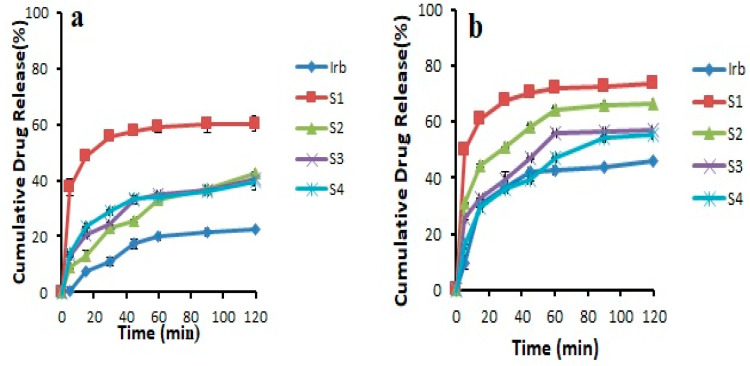
Dissolution profile of binary solid dispersions prepared with Soluplus^®^ in (**a**) pH 6.8 phosphate buffer (**b**) 0.1 N HCl (n = 3, mean ± SD).

**Figure 5 pharmaceutics-14-02264-f005:**
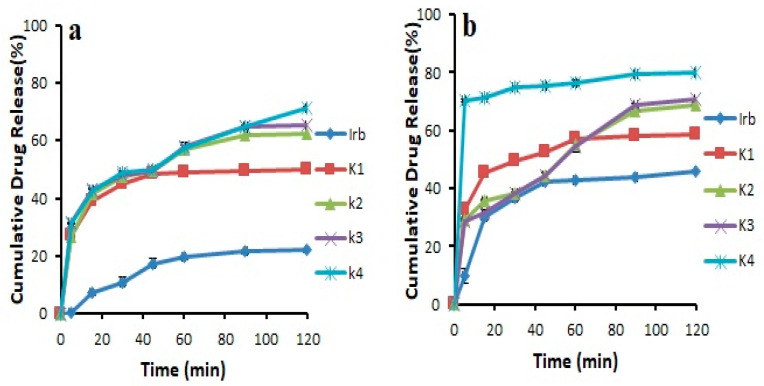
Dissolution profile of pure Irbesartan and binary solid dispersions prepared with Kollidon^®^ VA 64 in (**a**) pH 6.8 phosphate buffer (**b**) 0.1 N HCl (n = 3, mean ± SD).

**Figure 6 pharmaceutics-14-02264-f006:**
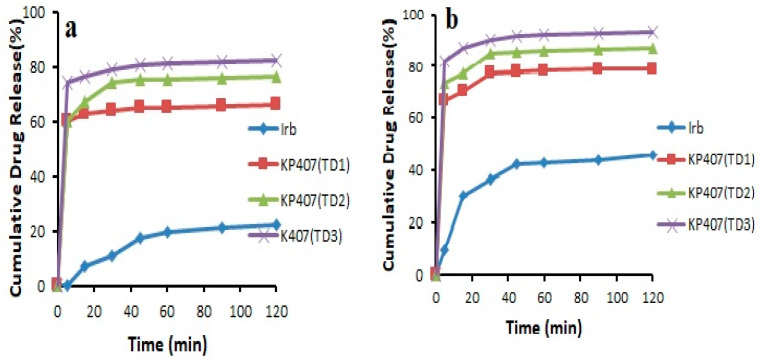
Dissolution profile of pure Irbesartan and ternary solid dispersions prepared with Kolliphor^®^ P 407 in (**a**) pH 6.8 phosphate buffer (**b**) 0.1 N HCl (n = 3, mean ± SD).

**Figure 7 pharmaceutics-14-02264-f007:**
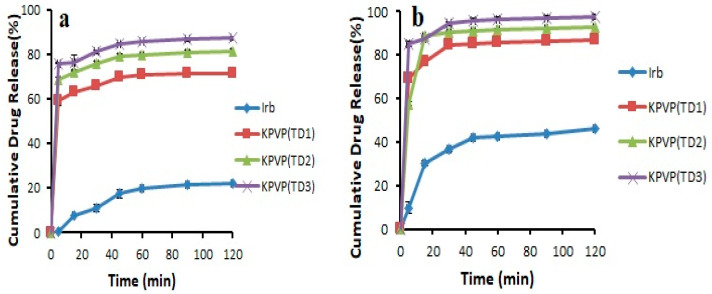
Dissolution profile of pure Irbesartan and ternary solid dispersions formulated with PVP- K30 in (**a**) pH 6.8 phosphate buffer (**b**) 0.1 N HCl (n = 3, mean ± SD).

**Figure 8 pharmaceutics-14-02264-f008:**
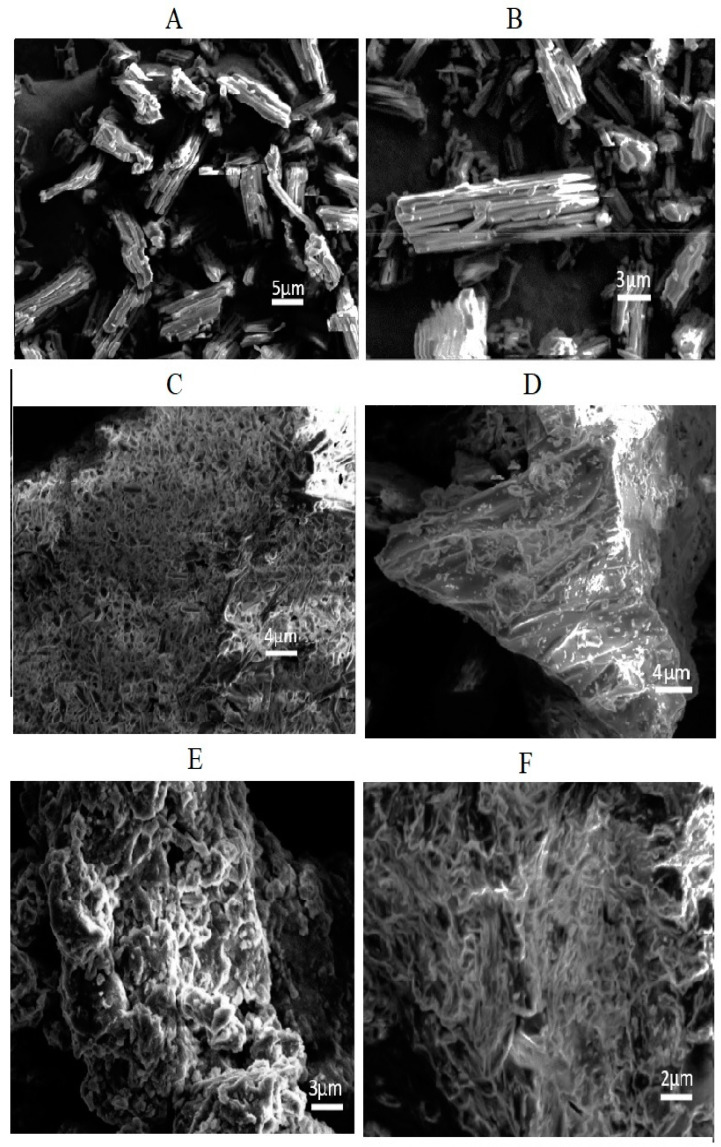
SEM micrographs of (**A**,**B**) pure Irbesartan, (**C**) S_1,_ (**D**) K_4,_ (**E**) KP407 (TD_3_), and (**F**) KPVP (TD_3_) formulations.

**Figure 9 pharmaceutics-14-02264-f009:**
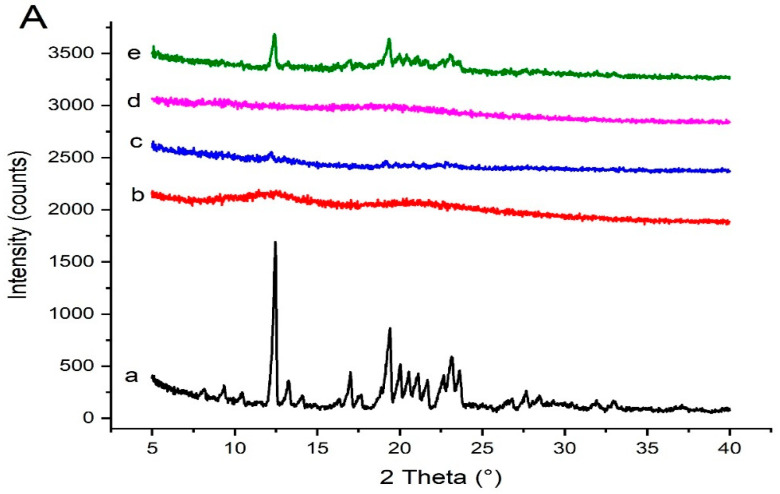
PXRD data of (**A**) Irbesartan and binary solid dispersions: (a) pure Irbesartan, (b) Kollidon^®^ VA 64, (c) binary solid dispersion of drug and Kollidon^®^ VA 64 at ratio 1:4 [K_4_], (d) Soluplus*^®^,* (e) binary dispersion of drug and Soluplus^®^ at ratio 1:1 [S_1_]. (**B**) Irbesartan and ternary solid dispersions: (a) pure Irbesartan, (b) Kollidon^®^ VA 64, (c) Kolliphor^®^ P 407, (d) Polyvinylpyrrolide- K30, (e) ternary solid dispersion KP407 (TD_3_), (f) ternary solid dispersion KPVP (TD_3_).

**Figure 10 pharmaceutics-14-02264-f010:**
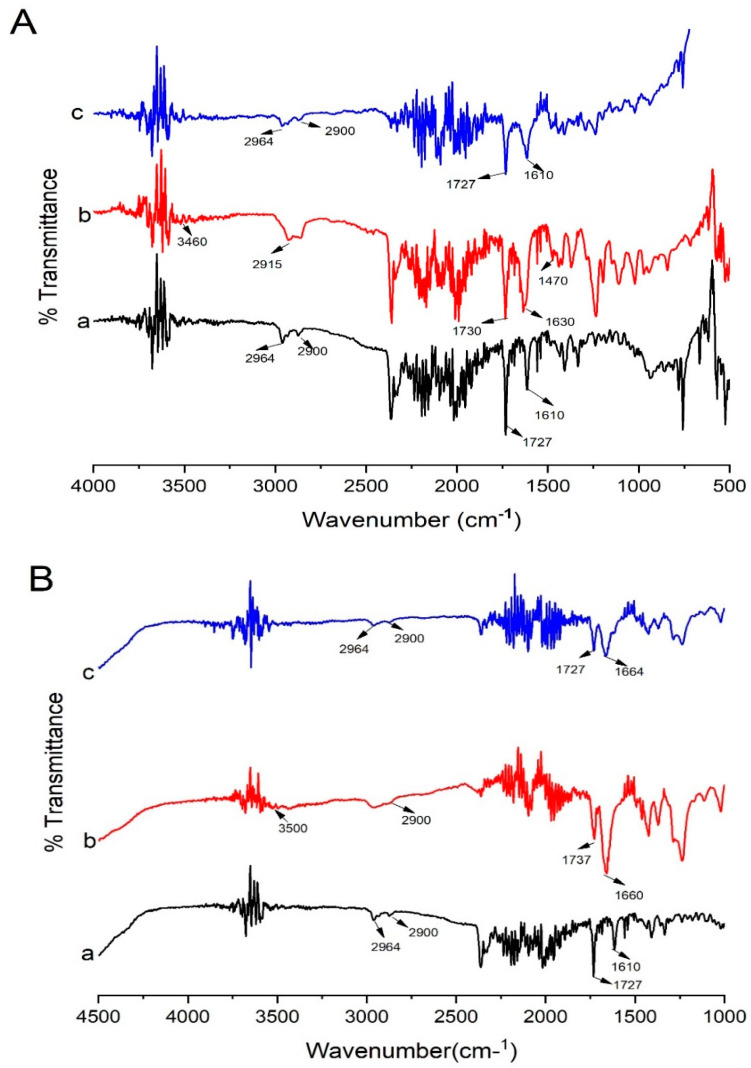
FTIR data of (**A**) Soluplus^®^-based solid dispersion formulation: (a) pure Irbesartan, (b) Soluplus*^®^,* (c) binary system of drug and Soluplus^®^ at ratio 1:1 [S_1_]. (**B**) Kollidon^®^ VA 64-based solid dispersion formulation: (a) pure Irbesartan, (b) Kollidon^®^ VA 64, (c) binary solid dispersion of drug and Kollidon^®^ VA 64 at ratio 1:4 [K_4_]. Note: Background noise may be present along with signals of compounds in FTIR display.

**Figure 11 pharmaceutics-14-02264-f011:**
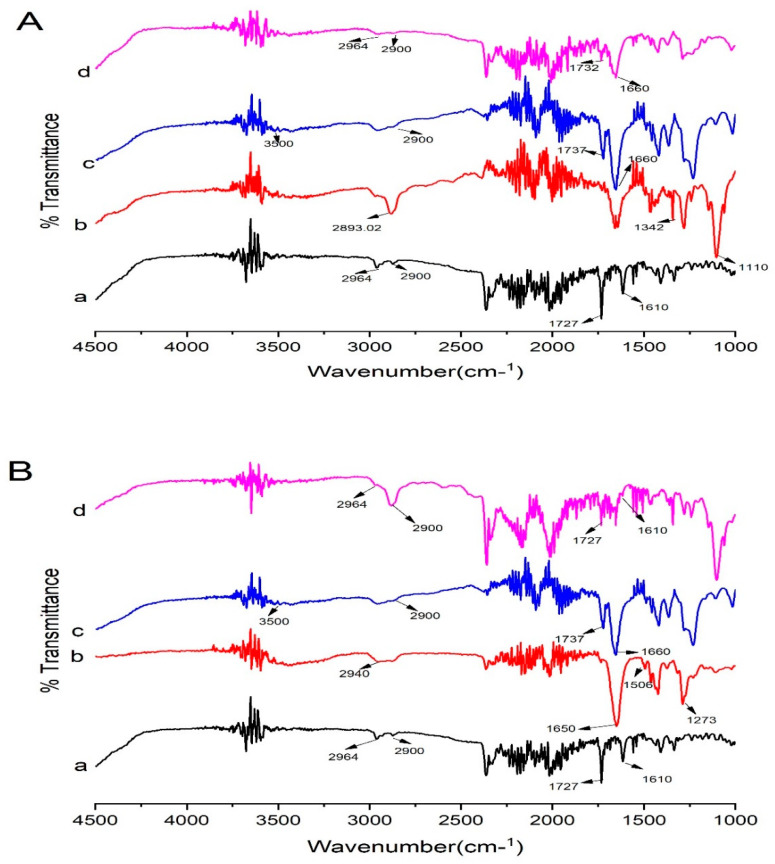
FTIR data of (**A**) Kolliphor^®^ P 407-based ternary solid dispersions formulation: (a) pure Irbesartan, (b) Kolliphor^®^ P 407, (c) Kollidon^®^ VA 64, (d) ternary system KP407 (TD_3_). (**B**) Polyvinylpyrrolidone-K30-based ternary solid dispersions formulation: (a) pure Irbesartan, (b) Polyvinylpyrrolidone-K30, (c) Kollidon^®^ VA 64, (d) ternary solid dispersion KPVP (TD_3_). Note: Background noise may be present along with signals of compounds in FTIR display.

**Figure 12 pharmaceutics-14-02264-f012:**
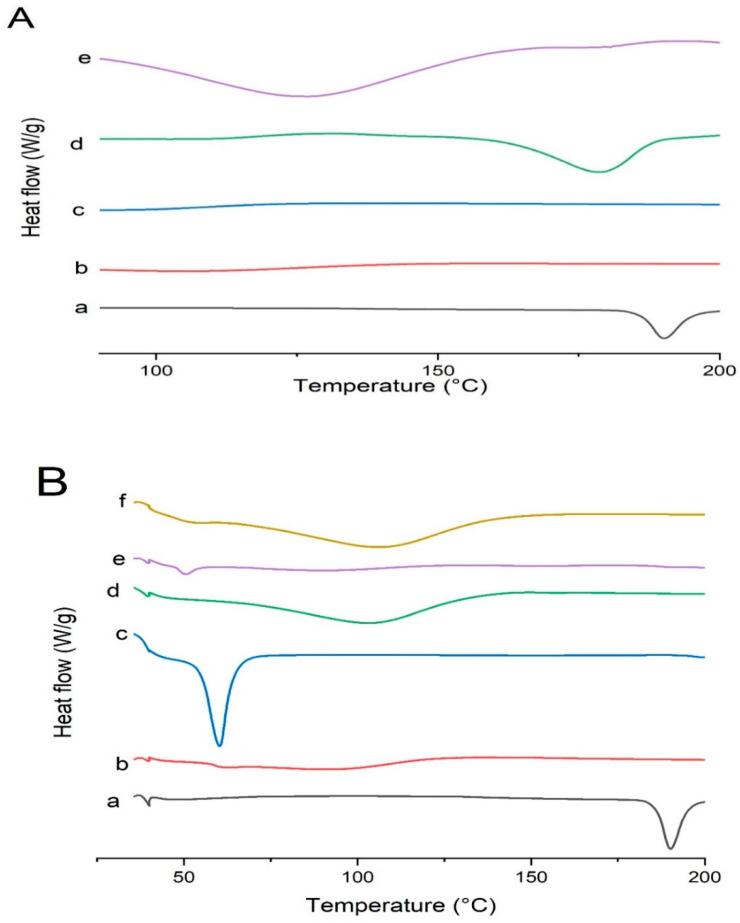
DSC curves of (**A**) Irbesartan and binary solid dispersions: (a) pure Irbesartan, (b) Soluplus*^®^,* (c) Kollidon^®^ VA 64, (d) binary dispersion of drug and Soluplus^®^ at ratio 1:1 [S_1_], (e) binary solid dispersion of drug and Kollidon^®^ VA 64 at ratio 1:4 [K_4_]. (**B**) Irbesartan and ternary solid dispersions: (a) pure Irbesartan, (b) Kollidon^®^ VA 64, (c*)* Kolliphor^®^ P 407, (d) Polyvinylpyrrolide- K30 (PVP-K30), (e) ternary solid dispersion KP407 (TD_3_), (f) ternary solid dispersion KPVP (TD_3_).

**Table 1 pharmaceutics-14-02264-t001:** Composition of binary and ternary solid dispersions.

Formulation Code (Binary Solid Dispersions)	Composition (Drug: Polymer), *w/w*
S1	1:1
S2	1:2
S3	1:3
S4	1:4
K1	1:1
K2	1:2
K3	1:3
K4	1:4
**Formulation code (Ternary solid dispersions)**	**Composition (Drug: Kollidon® VA 64: ternary polymer ), *w/w/w***
KP407 (TD1)	1:4:1
KP407 (TD2)	1:4:2
KP407 (TD3)	1:4:3
KPVP (TD1)	1:4:1
KPVP (TD2)	1:4:2
KPVP (TD3)	1:4:3

**Table 2 pharmaceutics-14-02264-t002:** Relationship of n value with release mechanism.

N	Release Mechanism
0.45 or less	Fickian diffusion
0.45–0.89	Non Fickian or anomalous mechanism
0.89	Case II transport
>0.89	Super case II transport

**Table 3 pharmaceutics-14-02264-t003:** Screening data of Irbesartan with different polymers at various concentrations.

Carrier Concentration, %	1a	2a	3a	4a
Soluplus^®^	162.24 ± 1.87	106.16 ± 1.87	67.53 ± 2.85	59.43 ± 1.87
Kollidon^®^ VA 64	170.34 ± 1.079	190.90 ± 1.079	230.15 ± 1.079	282.49 ± 2.15
PVP-K30	110.52 ± 5.39	115.51 ± 1.86	134.20 ± 1.86	172.21 ± 2.85
Kolliphor^®^ P 407	96.19 ± 3.89	109.90 ± 4.94	120.49 ± 4.31	169.71 ± 1.87
PEG 6000	70.03 ± 5.39	23.30 ± 3.89	25.79 ± 1.87	12.71 ± 1.87
HPMC E5	33.89 ± 4.70	50.09 ± 1.86	61.30 ± 1.86	77.50 ± 1.079
Gelucire^®^ 50/13	30.77 ± 2.15	27.04 ± 1.079	------- b	-------- b

a: Solubility (μg/mL) in respective carrier concentration. b: Insoluble in water.

**Table 4 pharmaceutics-14-02264-t004:** Drug content and % yield data of binary and ternary solid dispersion formulations.

Formulation	Drug Content, %	Yield, %
S1	92.89	92.00
S2	82.03	82.11
S3	87.88	84.55
S4	81.20	85.00
K1	86.21	90.00
K2	90.39	87.00
K3	86.21	88.00
K4	93.73	94.67
KP407 (TD1)	91.22	88.22
KP407 (TD2)	93.73	90.00
KP407 (TD3)	94.56	94.00
KPVP (TD1)	90.39	85.00
KPVP (TD2)	92.89	90.00
KPVP (TD3)	94.56	94.00

**Table 5 pharmaceutics-14-02264-t005:** Kinetic modelling data of prepared solid dispersion formulations from drug release in pH 6.8 phosphate buffer.

Formulaion Codes	Zero Order	First Order	Higuchi Model	Hixon-Crowell	Krosmeyer–Peppas	
	(R2)	(R2)	(R2)	(R2)	(R2)	n
S1	0.44	0.29	0.61	0.10	1.00	0.18
S2	0.53	0.73	0.97	0.67	1.00	0.39
S3	0.43	0.63	0.94	0.57	0.99	0.36
S4	0.16	0.41	0.86	0.34	0.99	0.29
K1	0.70	0.04	0.57	0.13	0.97	0.17
K2	0.03	0.51	0.79	0.36	0.99	0.25
K3	0.10	0.46	0.76	0.31	0.99	0.23
K4	0.08	0.57	0.83	0.45	1.00	0.04
KP407 (TD1)	0.51	0.33	0.03	0.67	1.00	0.04
KP407 (TD2)	0.51	0.34	0.14	0.04	0.99	0.07
KP407 (TD3)	0.51	0.60	0.03	0.01	1.00	0.04
KPVP (TD1)	0.56	0.02	0.09	0.25	1.00	0.06
KPVP (TD2)	0.55	0.50	0.07	0.06	1.00	0.06
KPVP (TD3)	0.54	0.74	0.03	0.10	1.00	0.05

**Table 6 pharmaceutics-14-02264-t006:** Kinetic modelling data of prepared solid dispersion formulations from drug release data in 0.1 N HCl.

Formulaion Codes	Zero Order	First Order	Higuchi Model	Hixon-Crowell	Krosmeyer–Peppas	
	(R2)	(R2)	(R2)	(R2)	(R2)	n
S1	0.92	0.30	0.37	0.10	0.99	0.12
S2	0.17	0.50	0.74	0.33	0.99	0.22
S3	0.11	0.54	0.84	0.42	0.98	0.27
S4	0.89	0.70	0.94	0.62	0.99	0.35
K1	0.56	0.13	0.56	0.07	0.99	0.16
K2	0.38	0.71	0.91	0.63	0.97	0.33
K3	0.50	0.78	0.93	0.72	0.97	0.37
K4	0.54	0.33	0.02	0.73	1.00	0.04
KP407 (TD1)	0.67	0.34	0.71	0.74	1.00	0.05
KP407 (TD2)	0.54	0.74	0.06	0.15	1.00	0.05
KP407 (TD3)	0.51	0.93	0.71	0.08	1.00	0.04
KPVP (TD1)	0.56	0.58	0.13	0.22	1.00	0.07
KPVP (TD2)	0.48	0.93	0.12	0.70	1.00	0.02
KPVP (TD3)	0.52	0.96	0.00	0.11	1.00	0.04

## Data Availability

The data presented in this study are available on request from the corresponding author. The data are not publicly available as it was originally produced through research.

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
