# Peer review of "How to Improve Solubility and Dissolution of Irbesartan by Fabricating Ternary Solid Dispersions: Optimization and In-Vitro Characterization"

_pharmaceutics, 2022, doi:10.3390/pharmaceutics14112264_

Round 1

Reviewer 1 Report

The manuscript deals with an optimization study with the aim to improve the solubility and dissolution features of the irbesartan drug.

Four polymer systems have been mixed and the best solution was proposed after the analytical evaluation. The approaches used in the research work are well known and widely used, as reported in the literature. However, the results seem interesting.

The manuscript could be improved on some points.

Comments:

Abstract - (Irbesartan)) - too many brackets, maybe it could be better without anyone.

- k4 - If readers see the abstract before the discussion, they will have some difficulty understanding which experiment it refers to.

- The aim of the work is to develop and optimize a solid dispersion to increase the solubility of irbesartan. The authors proceed with the first experimentation with 2 components and then add another 2 polymers. Did the authors evaluate the opportunity to adopt a Design of Experiment (DoE) approach with a simultaneous variation of all the components in the screening mode and therefore for the optimization of the mixture?

Author Response

Dear Reviewer

Please find the attached file addressing all changes as per your kind suggestions.

Best Regards,

Reviewer 2 Report

The manuscript “How to improve solubility and dissolution of Irbesartan by fabricating ternary solid dispersion: Optimization and in-vitro characterization” deals with the formulation of irbesartan, molecule well known for its poor solubility, in order to get it in amorphous and thus improve its solubility. A few common excipients were employed, also as mixtures, in the preparations of solid dispersion that led to the increase of the solubility of irbesartan. The successful amorphization of the drug is supported by the XRD, DSC and SEM observations.

In my opinion this piece of work presents a few matters of concern:

-FTIR spectroscopic section

the spectra show strong background signals (in particular CO2 stretching), a strong noise in the range2300-1800  cm-1 and above 3500 cm-1, like subtraction artefacts. Maybe the spectra were acquired using a blank KBr pellet while recording the background or there were instrumental problems.

The assignement of the signals of irbesartan do not fully agree with literature Franca, C.A., Etcheverry, S.B., Pis Diez, R. and Williams, P.A.M. (2009), Irbesartan: FTIR and Raman spectra. Density functional study on vibrational and NMR spectra. J. Raman Spectrosc., 40: 1296-1300. https://doi.org/10.1002/jrs.2282

 I suggest to delete the whole section because it does not provide immediate information about the interaction between the polymeric carrier and the drug and its form. Otherwise the authors should repeat all the measurements more accurately.

-Solubility studies

the solubility values for the pure drug in distilled water, pH6.8 and 0.1 N HCl  (10.84 ug/mL, 19.78 ug/mL and 30.72 ug/mL) and do not compare well with literature data (14.6 ug/mL, 115 ug/mL,  0.88 mg/mL ref17), the author should cite explicitly literature data and comment about it. The solubility in the strong acid is certainly much higher than in the other media due to the base properties of the molecules.

A matter of concern is the method used to determine the concentration because if the solubility increases very much the absorbance of the sample may exceed  the maximum value of  absorbance measurable by the spectrophotometer, usually 3. According to the Lambert-Beer law the absorbance is proportional to concentration. Which concentration range was used for calibration? Were the most concentrated solutions further diluted?

-Materials and methods

Missing information, i.e., the characteristics of the instruments and in the description of the  procedures. Ref. 17 can be taken as an example to be inspired.

Part of the text relevant to SEM  is at the end of the DSC section

 -PXRD studies

From the XRD powder pattern it can be inferred that the present sample corresponds to the crystalline A form

-Of minor importance the uneven formatting and some typesetting mistakes, e.g.,

Polymer nouns sometimes beginning with the capital letter, sometimes not, as well as not always italicized

Missing subscript and superscript in the equations of the Kinetic model analysis section

lines 183 and 185 Korsmeyer and Peppas are a surnames

italics in lines 71, 73, 74, 269-271

ref 71: Missing page number

line 102: better “concentrations” than “strength”

line 103: “of these solutes” instead of “of these solutions”

line 276: “poloxamer”, but is another commercial name of Kolliphor

line 373: “alteration in hydrophobicity of drug” is questionable since the molecule does not change, rather it is more probable the formation of a polymer drug complex

line 473 : more appropriate: “two hydrogen bond acceptor groups”

Author Response

(The authors gave the same response as above.)

Reviewer 3 Report

   The study provided a systematic approach to evaluate the optimal solid dispersion for Irbesartan based on saturation solubility, dissolution studie,DSC, SEM, XRD and FTIR studies. The KPVP (TD3) showed highest increase in solubility as well as dissolution rate among ternary solid dispersion formulations. The articles was done in enough to select the optimal polymer for the preparation of solid dispersion. However, major problems needed to be solved before acceptance for publication.

(1) In the introduction part (line 70-77) , more detail information should be given on the Polymers, such as the hydrogen bond acceptors, molecular weight or biocompatible nature.

(2) The chemical structure of the different kinds of polymers and Irbesartan should be provided, therefore, the interaction between different polymers and Irbesartan was easier to be observed.

(3) In the line of 284-285: Based on the results of solubility and dissolution studies, KPVP (TD3) ternary solid dispersion formulation was selected as the optimized formulation and was used for further studies. Here, the best solid dispersion for Irbesartan were selected. However, the characterization study of other solid dispersion was performed. I think it was not necessary. Therefore, I advised the authors to move the 3.2 and 3.3 part behind the DSC study.

(4) What is the supersaturation in the solid dispersion. And what it mean?

(5) In the 3.5, the release mechanisms of solid dispersion formulations were poor in discussion. Different release behaviors in different polymers and in different medium (HCl and pH 6.8 phosphate buffer). What is the meaning of different n values in the solid dispersions. This part must be clearly demonstrated.

(6) The title should be re-written. Except for the solubility and dissolution study, the articles also emphasized other parts.

(7) The SEM image was not clear.

(8) Polarize light microscope study was necessary to demonstrate the crystallinity or the amorphous of the drug in different solid dispersions..

(9) The storage stability of Irbesartan in the KPVP (TD3) must be given.

Author Response

(The authors gave the same response as above.)

Reviewer 4 Report

This paper documented a comprehensive work for enhanced solubility and dissolution of Irbesartan by SD. The major concern of this work is the novelty. SD has been widely studied and applied in both research and pharma industry, and the increased solubility and dissolution are expected. What are the challenges and contributions of this work to the field?

Other comments,

1) Please check the English of the manuscript. A few typos were found, for example, "Polyinylpyrrolidone" in Line 21, and "sloid" in Line 143.

2) Please add a full definition of GIT when it is first introduced in the manuscript.

3) It may be better to move the second paragraph in Section 2.11 to Section 2.8.

4) Please justify why Soluplus was selected in Section 3.1. The increase in Soluplus concentration resulted in decreased solubility. PVP-K30 showed good solubility as well, while it's not selected. In addition, The resolution of Table 3 is poor.

5) Please check the resolution of the Figures and Tables. The resolution of SEM images are very poor.

6) Please check the format of the manuscript. 

Author Response

(The authors gave the same response as above.)

Round 2

Reviewer 2 Report

In the revised version of the manuscript the missing information about the used instrumentation was introduced and the mentioned minor issues were addressed properly. I understand that it is difficult to replace the noisy IR spectra with accurate ones. The authors explained that it is due to the fact they had the IR spectra registered by another lab. In order to show that they did not disregarded the noise in the spectra, I suggest to add at least a note to the captions to the figures 10 and11 stating that the iR spectra display, beside the signals of the compounds, strong noise made of sharp signals. I also invite the author in the future to give their samples to another IR lab for measurements.

Check the reference list, e.g.,

refs 19  and 51: missing either page or article number

ref 32 author and title only

ref 49 missing article ID

Author Response

Please find the attached file addressing your kind suggestions.

Best Regards

Reviewer 3 Report

The authors addressed my comments seriously and I believe that the quality of manuscript is significantly approved. The work can be accepted.

Author Response

Dear Reviewer,

The English of the manuscript has been proof read by a English Professor and suggested points have been fine tuned.

Best Regards,

Reviewer 4 Report

Accept.

Author Response

The manuscript has been proof read once again and minor mistakes have been corrected.

Best Regards,